# Differential Anti-Tumor Effects of IFN-Inducible Chemokines CXCL9, CXCL10, and CXCL11 on a Mouse Squamous Cell Carcinoma Cell Line

**DOI:** 10.3390/medsci11020031

**Published:** 2023-04-25

**Authors:** Ari Matsumoto, Miki Hiroi, Kazumasa Mori, Nobuharu Yamamoto, Yoshihiro Ohmori

**Affiliations:** 1Division of Oral and Maxillofacial Surgery, Department of Diagnostic and Therapeutic Sciences, Meikai University School of Dentistry, 1-1 Keyakidai, Sakado 350-0283, Japan; 2Division of Basic Biology, Department of Oral Biology and Tissue Engineering, Meikai University School of Dentistry, 1-1 Keyakidai, Sakado 350-0283, Japan; 3Division of Microbiology and Immunology, Department of Oral Biology and Tissue Engineering, Meikai University School of Dentistry, 1-1 Keyakidai, Sakado 350-0283, Japan

**Keywords:** IFN-inducible chemokine, CXCL9, CXCL10, CXCL11, anti-tumor effect, dipeptidyl peptidase 4, DPP4, CD26

## Abstract

Chemokines are a group of cytokines involved in the mobilization of leukocytes, which play a role in host defense and a variety of pathological conditions, including cancer. Interferon (IFN)-inducible chemokines C-X-C motif ligand 9 (CXCL), CXCL10, and CXCL11 are anti-tumor chemokines; however, the differential anti-tumor effects of IFN-inducible chemokines are not completely understood. In this study, we investigated the anti-tumor effects of IFN-inducible chemokines by transferring chemokine expression vectors into a mouse squamous cell carcinoma cell line, SCCVII, to generate a cell line stably expressing chemokines and transplanted it into nude mice. The results showed that CXCL9- and CXCL11-expressing cells markedly inhibited tumor growth, whereas CXCL10-expressing cells did not inhibit growth. The NH_2_-terminal amino acid sequence of mouse CXCL10 contains a cleavage sequence by dipeptidyl peptidase 4 (DPP4), an enzyme that cleaves the peptide chain of chemokines. IHC staining indicated DPP4 expression in the stromal tissue, suggesting CXCL10 inactivation. These results suggest that the anti-tumor effects of IFN-inducible chemokines are affected by the expression of chemokine-cleaving enzymes in tumor tissues.

## 1. Introduction

Chemokines are a group of cytokines with molecular weights of 8–10 kDa, possessing chemotactic properties that mobilize immunocompetent cells [1]. Chemokines are structurally classified into four groups: CXC, CC, C, and CX3C, based on the position of the first two cysteine residues adjacent to the NH_2_-terminus of the chemokine protein [1,2]. More than 50 chemokines have been identified thus far, and they are responsible for the chemotactic migration and activation of immunocompetent cells, such as granulocytes, lymphocytes, and monocytes/macrophages that possess their respective chemokine receptors [1,2].

Interferon (IFN)-inducible chemokine C-X-C motif ligand 9 (CXCL9; also known as monokine induced by gamma interferon [Mig]) [3], CXCL10 (also known as interferon-inducible protein 10 [IP-10]) [4,5,6], and CXCL11 (also known as interferon-inducible T cell alpha chemoattractant [I-TAC]) [7,8] were identified as genes induced by IFN or bacterial lipopolysaccharide. The receptor for these IFN-inducible chemokines is CXCR3, which has three spliced isoforms, CXCR3A, CXCR3B, and CXCR3-alt, and CXCR3 differs in the expressed cells and biological activity induced by ligand binding [9]. CXCR3A is expressed on CD8^+^ cytotoxic T lymphocytes (CTLs), CD4^+^ helper type-1 T cells (Th1), and natural killer (NK) cells, and is mainly involved in the mobilization of these cells into inflammatory foci and tumor tissues [10,11]. In contrast, CXCR3B is expressed on vascular endothelial cells and mediates angiogenesis inhibition via ligand binding [12,13]. CXCR3-alt, a mutant generated by exon skipping, has been reported to bind only CXCL11 with low affinity and induce chemotaxis [14].

Analysis of tumor models in mice has shown that these IFN-inducible CXCR3 ligands exert anti-tumor effects by inducing infiltration of Th1, CTL, and NK cells and inhibiting angiogenesis [15,16,17,18,19]. In addition, studies on the relationship between the expression of IFN-inducible chemokines and prognosis in human solid tumors have shown that high expression of IFN-inducible chemokines in tumor tissues and serum is associated with favorable prognosis in patients with kidney cancer [20], cervical cancer [21], colorectal cancer [22,23], esophageal squamous cell carcinoma [24], and ovarian cancer [25].

On the other hand, a negative correlation between the higher expression level of these IFN-induced chemokines and higher tumor stage and a lower five-year survival rate has been observed in breast cancer [26,27], head and neck squamous cell carcinoma (HNSCC) [28,29,30], high-grade ovarian cancer [31], and lung cancer [32]. The biological activity of IFN-inducible chemokines is also affected by post-translational modifications [33,34]. CXCL10 acts as an antagonist when its NH_2_-terminal amino acid is cleaved by dipeptidyl peptidase 4 (DPP4)/CD26 [34,35], maintaining its binding to the CXCR3 receptor without inducing chemotactic signaling [36]. The antagonistic form of CXCL10 was detected in high-grade serous epithelial ovarian carcinomas [31]. It has also been shown that tumor cells express CXCR3, and these IFN-inducible chemokines increase their chemotactic ability and are involved in cell migration and metastasis [37,38,39]. Thus, IFN-inducible chemokines may exert anti-tumor effects depending on the site of tumorigenesis or may be involved in promoting tumor progression and metastasis; however, the factors responsible for their anti-tumor or tumor-promoting effects remain elusive.

In the present study, we examined the anti-tumor effects of the IFN-inducible chemokines CXCL9, CXCL10, and CXCL11 by generating stably expressing cell lines in which the genes for the three IFN-inducible chemokines were introduced into a mouse squamous cell carcinoma (SCC) cell line and transplanted into nude mice. The results indicate that although CXCL9- and CXCL11-expressing SCCs exert anti-tumor activity, the CXCL10-expressing SCC failed to exert anti-tumor activity. The differential anti-tumor activity of these IFN-inducible chemokines appears to depend on the penultimate amino acid sequence of the NH_2_-terminus of the chemokine peptide chain that is cleaved by DPP4/CD26 [34,35], which is expressed in the tumor stromal tissue.

## 2. Materials and Methods

### 2.1. Cell Culture

The mouse squamous cell carcinoma cell line, SCCⅦ, was obtained from the Cell Bank, Center for Medical Cell Resources, Institute of Development, Aging and Cancer, Tohoku University (Sendai, Japan). This cell line was established from a cutaneous squamous cell carcinoma derived from C3H/He mice [40] and has been widely used as a mouse oral squamous cell carcinoma model [41,42,43]. The cells were cultured in complete RPMI1640 (Thermo Fisher Scientific, Waltham, MA, USA) containing 10% fetal bovine serum (FBS; Bio West, Miami, FL, USA), 20 mM L-glutamine, and 1% penicillin G/streptomycin sulfate (Thermo Fisher Scientific, Waltham, MA, USA) at 37 °C in a 5% CO_2_ incubator and subcultured every three days. Cell passaging was performed by washing the cells with phosphate-buffered saline (PBS; Thermo Fisher Scientific, Waltham, MA, USA). Subsequently, a 0.25% trypsin/0.01% EDTA (trypsin/EDTA; Thermo Fisher Scientific, Waltham, MA, USA) solution was used to detach the cells from the culture dish in order to obtain single-cell suspensions. After washing with PBS, cell numbers were measured using a hemocytometer and adjusted to the specified cell number for the experiment. Furthermore, 293FT cells used for lentivirus production were obtained from Invitrogen (Grand Island, NY, USA). The cells were cultured in Dulbecco’s Modified Eagle’s Medium (DMEM; Thermo Fisher Scientific, Waltham, MA, USA) containing 10% FBS and 1% penicillin G/streptomycin sulfate at 37 °C in a 5% CO_2_ incubator.

### 2.2. Lentiviral Expression Vectors

The cDNA plasmids pCMV-SPORT6 for the mouse IFN-inducible chemokine genes Cxcl9 (NM_008599) [3], Cxcl10 (NM_021274) [5], and Cxcl11 (NM_019494) [8] were obtained from Open Biosystems (Huntsville, AL, USA). To construct a lentiviral expression system for these IFN-inducible chemokines, entry vectors were generated using a Gateway cloning kit (Invitrogen, Grand Island, NY, USA) [44]. Polymerase chain reaction was performed with the mouse IFN-inducible chemokine cDNA plasmid as a template to amplify the translated region of the IFN-inducible gene using oligo DNA primers with *attB1* and *attB2* sequences (Appendix A). The amplified translation region was cloned into the entry vector (pDONR221, Invitrogen) and then into the expression vector pLent6/V5-DEST (Invitrogen, Grand Island, NY, USA) using LR clonase (Invitrogen, Grand Island, NY, USA), according to the manufacturer’s protocol. These expression vectors were sequenced using DNA sequencing (MacroGen, Tokyo, Japan) to confirm the nucleotide sequence. To prepare a lentiviral expression system, 6 × 10^6^ 293FT cells/dish were seeded in a 10-cm dish and transfected with the expression vector pLent6/V5-DEST and a plasmid encoding the structural protein of lentivirus (ViraPower Packaging Mix, Invitrogen, Grand Island, NY, USA) using Lipofectamine 2000 (Invitrogen, Grand Island, NY, USA). The culture supernatant containing virus particles was collected after 72 h.

### 2.3. Generation of Cell Lines Stably Expressing the IFN-Inducible Chemokines

Mouse SCCⅦ cells were seeded in 6-well plates (Thermo Fisher Scientific, Waltham, MA, USA) at a density of 2 × 10^4^ cells/well and cultured in a complete medium at 37 °C in a 5% CO_2_ incubator. After 24 h, the cells were incubated with 1 mL of lentivirus containing the IFN-inducible chemokine gene in the presence of polybrene (6 mg/well; Invitrogen, Grand Island, NY, USA). Forty-eight hours after transfection, blasticidin (10 mg/mL; Invitrogen, Grand Island, NY, USA) was added, and the cells were cultured for 24 h. The cells were then washed with PBS, detached using trypsin/EDTA, and single-cell suspensions were seeded into a 10-cm dish. After six days of culture in the presence of blasticidin, drug-resistant clones were detached using trypsin/EDTA, seeded at 1 cell/well in 96-well plates (Thermo Fisher Scientific, Waltham, MA, USA), and inoculated for 13 days. After culturing, the growing drug-resistant cells were transferred to 6-well plates and maintained in the presence of blasticidin.

To examine whether these drug-resistant clones produced IFN-inducible chemokines, 5–7 clones were isolated and seeded onto 24-well plates (Thermo Fisher Scientific, Waltham, MA, USA) at a density of 1 × 10^5^ cells/mL. The culture supernatant was collected after 72 h, and chemokine expression levels were determined using a Mouse Simple Step enzyme-linked immunosorbent assay (ELISA) kit (Abcam, Cambridge, UK) for CXCL9 (Mig), CXCL10 (IP-10), and CXCL11 (ITAC).

### 2.4. Cell Proliferation Assay

The IFN-inducible chemokine-expressing cells, empty vector-transfected cells (Vector), and the parental line SCCⅦ were seeded in 6-cm dishes at a density of 1 × 10^5^ cells in 10 mL of complete medium. After 24, 48, and 72 h, the cells were detached using trypsin/EDTA, and cell counts were measured using a hemocytometer (Erma, Tokyo, Japan).

### 2.5. Transplantation of a Cell Line Stably Expressing IFN-Inducible Chemokines into Nude Mice

Stable chemokine-expressing cells (n = 5 each), vector-transduced cells (n = 5), and the parental line SCCⅦ (n = 5) were subcutaneously transplanted into the back of 8–10-week-old BALB/cSlc-nu/nu mice (Sankyo Lab Service, Tokyo, Japan) at 9.5 × 10^6^ cells/200 μL using a syringe equipped with a 29-gauge injection needle (Terumo Myjector, Terumo, Tokyo, Japan) under inhalation anesthesia (Escain inhalation anesthetic solution, Pfizer, Tokyo, Japan). The tumor size was determined by measuring the long and short diameters using calipers. Tumor size was calculated using the following formula:Tumor volume (mm^3^) = Long diameter × (Short diameter × Short diameter)/2

Three weeks after the tumors in the Vector group reached their maximum size, all mice were euthanized by over-anesthetization with isoflurane (Escain inhalation anesthetic solution). The tumor was removed in its entirety as a single mass, including the epidermis, for gross observation and size measurement. The extracted tumors were fixed in 10% neutral buffered formalin solution and embedded in paraffin for immunohistochemical (IHC) studies. The study proposal was submitted to the Ethics Committee for Laboratory Animals of the Meikai University School of Dentistry and approved on 20 December 2018 in accordance with the guidelines for Institutional Laboratory Animals Care (approval code: A1843).

### 2.6. IHC Analysis

Paraffin-embedded tissues were cut into 4-μm slices and mounted onto poly-L-lysine coated slides (New Silane II, Mutoh Chemical, Tokyo, Japan). The tissue sections were deparaffinized by immersing in xylene, rehydrated in decreasing concentrations of alcohol, and incubated in a citrate buffer (pH 6.0). For antigen retrieval, the sections were heated in a microwave for 30 min in a citrate buffer. After cooling, the specimens were washed with Tris-buffered saline (TBS [pH 7.4]) and incubated with Dako REALTM peroxidase-blocking solution (Dako, Carpinteria, CA, USA) at 25 °C for 10 min to block endogenous peroxidase activity. To prevent non-specific reactions with endogenous mouse immunoglobulin, blocking with a Histofine Universal Reagent Mouse Stain Kit (Nichirei Biosciences, Tokyo, Japan) was performed at 25 °C for 60 min. The tissue sections were then incubated with goat anti-mouse DPP4 (CD26) polyclonal antibody (AF954; R&D Systems, Minneapolis, MN, USA) at 4 °C in a humidified chamber for 16 h, washed with TBS for 30 min, and blocked with a Histofine Universal Reagent Mouse Stain Kit (Nichirei) at 25 °C for 30 min. The tissue sections were washed in TBS and incubated with horseradish peroxidase-conjugated simple stain mouse MAX-PO (Nichirei Bioscience) at 25 °C for 30 min. Peroxidase activity was visualized by immersing the tissue sections in 3,3′-diaminobenzidine tetrahydrochloride. Finally, the tissue sections were counterstained with Mayer’s hematoxylin and then mounted.

### 2.7. Statistical Analyses

One-way analysis of variance (ANOVA) for multiple data was used to test for statistically significant differences using GraphPad Prism9 (version 9.1.0, GraphPad Software, San Diego, CA, USA). After confirming significance using one-way ANOVA, Dunnett’s method was used for multiple comparison tests with the Vector group. A *p*-value < 0.05 was considered statistically significant.

## 3. Results

### 3.1. Generation of Cell Lines Stably Expressing IFN-Inducible Chemokines and Investigation of Their Proliferation In Vitro

To investigate the anti-tumor effects of the IFN-inducible chemokines CXCL9, CXCL10, and CXCL11, we generated a cell line stably expressing these IFN-inducible chemokines using the mouse squamous cell carcinoma cell line SCCⅦ. Blasticidin-resistant cells transfected with only the expression vector were used as controls. The chemokine production ability of the drug-resistant cell lines was examined using ELISA after 72 h of culture (Figure 1). The results showed that cell lines transfected with each chemokine expression vector produced significant amounts of that chemokine (*p* < 0.001).

To examine the proliferative capacity of the stable chemokine-expressing cell lines in vitro, the parental SCCⅦ cells, Vector, and the stably expressing cell lines were seeded in 6-cm dishes, and the cell numbers were measured over time (Figure 2). The parental SCCⅦ cells showed a marked increase in cell number during 24–48 h of culture, whereas the stable chemokine-expressing cell lines showed no significant difference in proliferative capacity compared to that of the control Vector.

### 3.2. Anti-Tumor Effects of IFN-Inducible Chemokines in Nude Mice

To investigate the anti-tumor effect of IFN-inducible chemokines in vivo, we transplanted chemokine-expressing cell lines into the back of nude mice and measured the tumor size on days 8, 11, 15, and 18 after transplantation (Figure 3). The CXCL9- and CXCL11-expressing cell lines significantly suppressed tumor growth on day 11 post-inoculation compared to that by the Vector, and this suppression was observed until day 18 (*p* < 0.05). In contrast, the CXCL10-expressing cell line did not inhibit tumor growth, and the growth was similar to that of the Vector. Three weeks after transplantation, tumors were extracted for gross observation and measurement of mass size (Appendix A). Ulcerations caused by an overgrowth of tumors were observed in tumors transplanted with parental SCCVII cells, Vector, and CXCL10-expressing cell lines, while such tissue distraction was not observed in the CXCL9- and CXCL11-expressing cell lines. The size of the extracted tumors was significantly smaller in the CXCL9- and CXCL11-expressing cell lines as compared to the CXCL10-expressing cell line. These results indicate that the anti-tumor effects of the IFN-inducible chemokines CXCL9, CXCL10, and CXCL11 differ in vivo.

### 3.3. IHC Study of DPP4 in Tumor Tissues Expressing the IFN-Inducible Chemokines

The biological activity of chemokines is related not only to the number of chemokines expressed in tissues and the expression of receptors on responding cells but also to the presence of enzymes that degrade chemokines [45]. DPP4, also known as CD26, has been shown to cleave the peptide chain of chemokines at the NH_2_-terminal, if a proline or alanine is present at the penultimate position of the NH_2_-terminal [34]. When the amino acid sequences of mouse IFN-inducible chemokines were inspected, only CXCL10 had a proline residue at the penultimate position of the NH_2_-terminus (Figure 4A).

To explore the cause of this difference in the anti-tumor effects of IFN-induced chemokines, we performed IHC studies on the expression of DPP4 in tumor tissues (Figure 5). DPP4-positive findings were observed in the subcutaneous tissues (stroma) of all transplanted tumor tissues, including the parental SCCⅦ cells.

These results indicate that the IFN-inducible CXCR3 ligands show differences in their anti-tumor effects in a mouse tumor model and suggest that the differences are related to the sensitivity to the chemokine-cleaving enzyme DPP4 expressed in tumor tissues.

## 4. Discussion

IFN-inducible chemokines, known as CXCR3 ligand chemokines [11], have been recognized as anti-tumor chemokines in mouse tumor models and clinicopathological analyses of human solid tumors [15,16,17,18,19,20,21,22,23,24,25]. In contrast, no anti-tumor effect has been observed in breast cancer [26,27], HNSCC [28,29,30], high-grade ovarian cancer [31], and lung cancer [32]. However, the reason behind this difference in the opposing biological activities of IFN-inducible chemokines remains elusive. In the present study, we constructed cells stably expressing the three IFN-inducible chemokines CXCL9, CXCL10, and CXCL11 in a mouse model and transplanted them into nude mice to investigate their anti-tumor effects. CXCL9- and CXCL11-expressing cells showed significant anti-tumor activity, while CXCL10-expressing cells failed to exert an anti-tumor effect. This difference in the anti-tumor effects of IFN-inducible chemokines was suggested to be dependent on the sensitivity to DPP4/CD26 [46,47], an enzyme that cleaves chemokines expressed in tumor tissues.

In this study, we constructed a cell line that stably expresses IFN-inducible chemokines using the SCCⅦ mouse squamous cell carcinoma cell line. This cell line was derived from mouse skin squamous cell carcinoma [40] and has been widely used in mouse oral squamous cell carcinoma models [41,42,43]. Some tumor cell lines express CXCR3, a receptor for IFN-inducible chemokines, and it has been reported that chemokines produced by tumor cells stimulate cell motility and are involved in tumor progression and metastasis [37,38,39]. The in vitro proliferative capacity of the stable chemokine-expressing cells derived from SCCⅦ cells established in this study was not significantly different from that of control cells transfected with the empty vector. Thus, the difference in tumor growth in vivo by the IFN-inducible chemokines is not due to the difference in the proliferative capacity of the established stable chemokine-expressing cells.

An important finding of this study was the involvement of DPP4 [34,48], a membrane-bound serine protease, in the differential anti-tumor effects of IFN-induced chemokines. DPP4 is expressed not only by immune cells, such as macrophages, B cells, T cells, and NK cells, but also by fibroblasts, endothelial cells, and epithelial cells [34,48]. DPP4 selectively cleaves dipeptides at the NH_2_-terminal of the protein, if proline or alanine is present at the penultimate position of the NH_2_-terminal [34]. Among the IFN-inducible chemokines, only mouse CXCL10 has a proline at the penultimate position (Figure 4A), which is a cleavage sequence of DPP4. Therefore, the difference in the anti-tumor effect of the IFN-inducible chemokines examined in this study is thought to be due to the difference in sensitivity to DPP4. IHC staining for DPP4 expression in tumor tissues showed that DPP4 was expressed in the stromal tissues of all nude mice implanted with tumor cells, suggesting that only CXCL10 produced by the implanted cells was inactivated by DPP4.

Recently, DPP4/CD26 was identified as a new marker for cancer-associated fibroblasts (CAFs), and its expression has been reported to increase with breast cancer progression [49]. The presence of DPP4-cleaved shortened CXCL10 in human high-grade ovarian cancer tissues suggests that the truncated form of CXCL10 may induce an immunosuppressive tumor microenvironment [31]. It has also been reported that inhibition of the enzymatic activity of DPP4 promotes infiltration of CXCR3-expressing T cells into tumor tissues and enhances their anti-tumor effects [50]. Since the NH_2_-terminal amino acid sequences of the human IFN-induced chemokines CXCL9, CXCL10, and CXCL11 all contain this DPP4 cleavage-sensitive sequence (Figure 4B), it is possible that DPP4 expressed in the tumor microenvironment has a significant impact on the differential anti-tumor effects of IFN-induced chemokines, as reported in previous studies [26,27,28,29,30,31,32].

The anti-tumor effects of IFN-inducible chemokines have been shown to depend on the infiltration of CD8^+^ CTL and NK cells [15,16,17,18,19] and their ability to suppress angiogenesis [12,13]. Since athymic nude mice lack T cells, we examined the expression of NK1.1 (CD161), a marker of NK cells [51], using IHC staining in the course of this study. However, no marked infiltration of NK1.1-positive cells was observed in CXCL9- and CXCL11-expressing tumor tissues (data not shown). In addition, we investigated whether the expression of angiogenic factors, such as vascular endothelial growth factor (VEGF), was suppressed in CXCL9- and CXCL11-expressing tumor tissues using IHC. Although higher expression of VEGF was observed in tumor tissues transplanted with parental SCCVII cells, no significant suppression of VEGF expression was observed in CXCL9- and CXCL11-expressing tumor tissues (data not shown). Further studies are required to determine the anti-tumor mechanisms of CXCL9- and CXCL11-expressing tumor tissues in nude mice.

## 5. Conclusions

In this study, we established SCCVII cells stably expressing the IFN-inducible chemokines CXCL9, CXCL10, and CXCL11 and transplanted the cells into the backs of nude mice to investigate their anti-tumor effects. The results demonstrated that although CXCL9- and CXCL11-expressing cells showed anti-tumor effects, CXCL10-expressing cells failed to exert anti-tumor effects. Additionally, the differential anti-tumor activity of IFN-inducible chemokines is dependent on their sensitivity to DPP4/CD26, which cleaves the dipeptide at the NH_2_-terminal of the chemokines. These findings indicate that the different anti-tumor activities of IFN-inducible chemokines depend on the expression of DPP4 in the tissues where tumors develop and on the differential sensitivity of chemokines to DPP4. Further studies are required to determine the anti-tumor mechanisms of CXCL9- and CXCL11-expressing tumor tissues, which are resistant to DPP4, to elucidate the immunosuppressive role of DPP4 in the tumor microenvironment, and to test whether DPP4 inhibitors restore the anti-tumor activity of IFN-inducible chemokines in an immunocompetent mouse model.

## Figures and Tables

**Figure 1 medsci-11-00031-f001:**
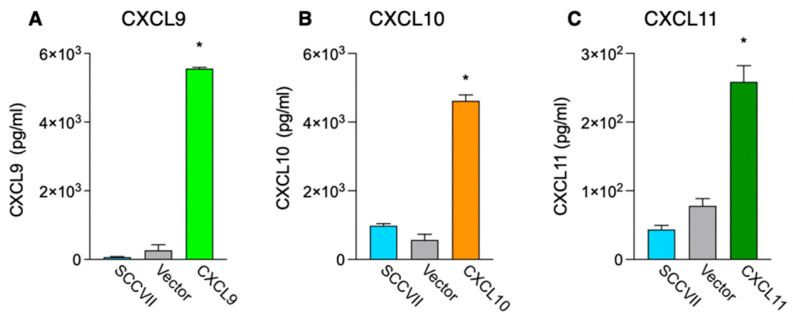
Production of CXCL9 (**A**), CXCL10 (**B**), and CXCL11 (**C**) from SCCVII cells transduced with the chemokine expression vector. The blasticidin-resistant chemokine expression vector-transfected, Vector, or parental SCCVII cells were cultured in DMEM supplemented with 10% FBS for 72 h, and the culture supernatants were then assessed for the amount of chemokine production by ELISA. Each column and bar represents the mean + SD (n = 3). Statistical differences in the amount of chemokine production relative to empty vector-transfected cells are indicated (* *p* < 0.001, one-way ANOVA).

**Figure 2 medsci-11-00031-f002:**
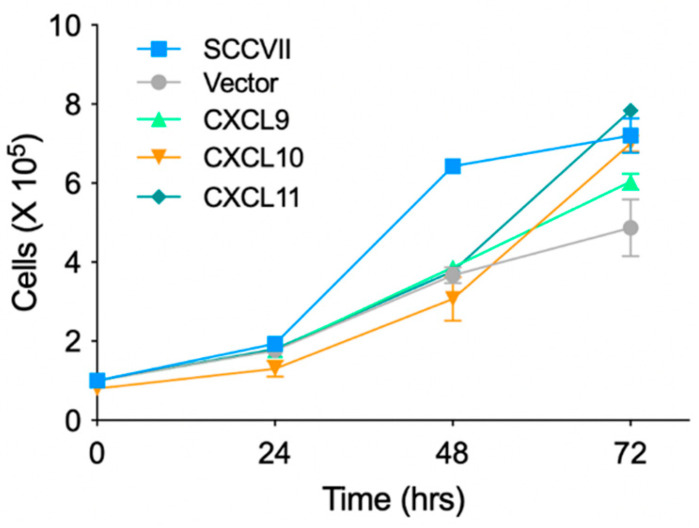
No significant difference in the growth rate in vitro among the stable cell lines expressing the IFN-inducible chemokines. Chemokine-expressing cells, Vector, or parental SCCVII cells were seeded in a 6-cm dish at a density of 1 × 10^5^ cells/dish and grown for the indicated time before the cell count was assessed. Each symbol and bar represents the mean ± SD (n = 3). No statistically significant differences were observed in the cell numbers of the chemokine-expressing cells compared to empty vector-transfected cells (one-way ANOVA).

**Figure 3 medsci-11-00031-f003:**
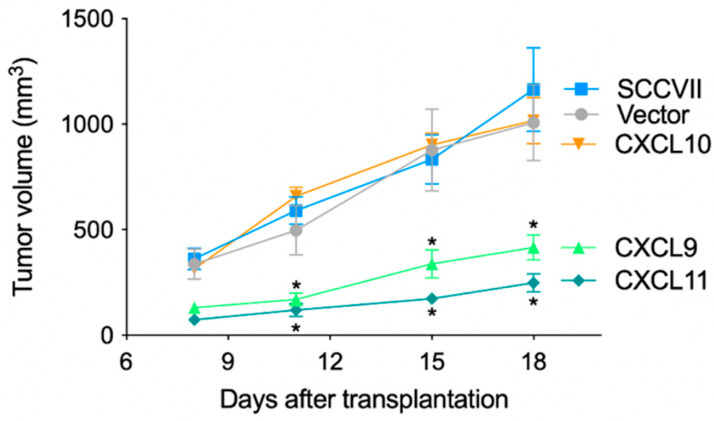
Differential anti-tumor effect of the transplanted chemokine-expressing cells in nude mice. Chemokine-expressing, Vector, or parental SCCVII cells were transplanted into the back of nude mice. Tumor growth was monitored by measuring the tumor size in two cross directions. Each symbol and bar represent the mean ± SD (n = 5). Statistical differences in tumor size relative to empty vector-transplanted cells are indicated (* *p* < 0.05, one-way ANOVA).

**Figure 4 medsci-11-00031-f004:**
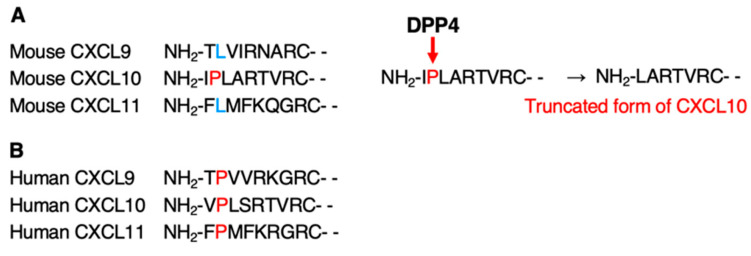
Amino acid sequences of the NH_2_-terminal region of the IFN-inducible chemokine and the cleavage site by DPP4. (**A**). DPP4, also known as CD26, is a serine protease capable of enzymatic removal of the first two amino acids from a protein that possesses proline (P) or alanine (A) in the penultimate NH_2_-terminal position. The first two amino acids of mouse CXCL10 are enzymatically cleaved by DPP4. The truncated form of CXCL10 loses its chemotactic activity and possibly functions as an antagonist that binds to its receptor, CXCR3 [31,34]. (**B**). Amino acid sequence of NH_2_-terminal region of the human IFN-inducible chemokine and the cleavage site by DPP4. These amino acid sequences were obtained from UniProt (https://www.uniprot.org; accessed on 25 September 2021).

**Figure 5 medsci-11-00031-f005:**
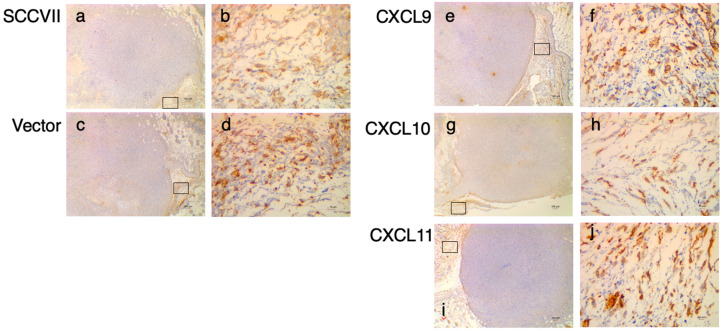
IHC analysis of dipeptidyl peptidase 4 (DPP4) expression in a nude mouse transplanted with chemokine-expressing cells. IHC analysis of DPP4 expression in tumor stroma in nude mice transplanted with chemokine-expressing cells. Original images were captured at 40× (**a**,**c**,**e**,**g**,**i**): scale bar = 100 μm), and the rectangular regions were captured at 400× in stroma (**b**,**d**,**f**,**h**,**j**): scale bar = 10 μm).

## Data Availability

The data presented in this study are available in this article and the Appendix A.

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
