# Peer review of "Differential Anti-Tumor Effects of IFN-Inducible Chemokines CXCL9, CXCL10, and CXCL11 on a Mouse Squamous Cell Carcinoma Cell Line"

_medsci, 2023, doi:10.3390/medsci11020031_

Round 1

Reviewer 1 Report

The manuscript entitled “Differential Anti-tumor Effects of IFN-inducible Chemokines 2 CXCL9, CXCL10, and CXCL11 on Mouse Squamous Cell 3 Carcinoma Cell Line” describes the difference in anti-tumor efficacy of three IFN-inducible chemokines CXCL9, CXCL10 and CXCL11 in mice and suggests critical function of chemokine cleaving enzymes in modulating the anti-tumor function of these chemokines.

The manuscript would be of great interest to readers and attempts to understand the mechanism of differential anti-tumor function of chemokines against mouse squamous cell carcinoma. Please find my comments below to improve the manuscript before publication in Medical Sciences.

Major-

1. Please explain what tissue sections were removed and used for IHC, it is alteast not clear from the method section.

2. Please mention the mode of injecting (i.p., i.v. etc) the chemokine expressing SCCVII into the nude mice. It was not clear in the methods section.

3. For cell proliferation assay, CFSE or Ki67 assay would have been more prominent to check the proliferation. Did authors ever consider it.

Author Response

Responses to Reviewer #1 Comments

  1. Please explain what tissue sections were removed and used for IHC, it is alteast not clear from the method section.

Reply: The tumor was removed in its entirety as a single mass, including the epidermis, and fixed in formalin after gross observation and size measurement. We have revised the text in the method section as follows (lines 158-163):

“Three weeks after the tumors in the Vector group had reached their maximum size, all mice were euthanized by over-anesthetization with isoflurane (Escain inhalation anes-thetic solution). The tumor was removed in its entirety as a single mass, including the epidermis, for gross observation and size measurement. The extracted tumors were fixed in 10% neutral buffered formalin solution and embedded in paraffin for immunohistochemical (IHC) studies.”

  1. Please mention the mode of injecting (i.p., i.v. etc) the chemokine expressing SCCVII into the nude mice. It was not clear in the methods section.

Reply: Chemokine expressing SCCVII cells were subcutaneously injected into the back of nude mice. We have revised the text as follows (line 149):

“Stable chemokine-expressing cells (n = 5 each), vector-transduced cells (n = 5), and the parental line SCCâ…¦ (n = 5) were subcutaneously transplanted into the back of 8–10-week-old BALB/cSlc-nu/nu mice (Sankyo Lab Service, Tokyo, Japan) at 9.5 × 106 cells/200 μL using a syringe equipped with a 29-gauge injection needle (Terumo Myjector, Terumo, Tokyo, Japan) under inhalation anesthesia (Escain inhalation anesthetic solution, Pfizer, Tokyo, Japan).”

  1. For cell proliferation assay, CFSE or Ki67 assay would have been more prominent to check the proliferation. Did authors ever consider it.

Reply: CFSE and Ki-67 are cellular markers for proliferation and have been widely used in flow cytometry and immunohistochemistry, respectively. In the present study, we directly and quantitatively measured the number of proliferating cells in vitro over time, instead of these proliferation markers.

Reviewer 2 Report

In this manuscript, the SCC cells stably expressing IFN-inducible chemokines CXCL9, CXCL10, and CXCL11 were established and transplanted into nude mice to investigate the anti-tumor effects. The results demonstrated that although CXCL9- and CXCL11-expressing cells showed anti-tumor effects. CXCL10-expressing cells failed to exert anti-tumor effects. Additionally, the differential anti-tumor activity of IFN-inducible chemokines is dependent of their sensitivity to DPP4/CD26, which cleaves the dipeptide at the NH2-terminal of the chemokines. The work is interesting and meaningful. However, the work is a little simple. If the following work is performed, the work can be considered for publication.

1. The authors claimed that SCC cells stably express chemokines CXCL9, CXCL10, and CXCL11. What is the expression level? The specific expression amount should be determined.

2. The SCC cells were transplanted into nude mice to investigate the antitumor effects. The tumor volume was quantified. How about the tumor mass?

3. The results are simple. The antitumor mechanism should be investigated and appended.

Author Response

Responses to Reviewer #2 Comments

  1. The authors claimed that SCC cells stably express chemokines CXCL9, CXCL10, and CXCL11. What is the expression level? The specific expression amount should be determined.

Reply: Thank you for your insightful question. We have already provided the data for the production amount of CXCL9, CXCL10, and CXCL11 from the SCCVII cells transduced with the chemokine expression vector using ELISA in Figure 1 (lines 200–202). Our results demonstrate that the SCCVII cells stably expressed each chemokine produced at significantly higher levels compared to vector-transduced cells.

  1. The SCC cells were transplanted into nude mice to investigate the antitumor effects. The tumor volume was quantified. How about the tumor mass?

Reply: Thank you for your insightful question. The volumes of tumors formed on the back of nude mice were measured for up to 18 days. On day 21, ulceration was observed on the epidermis at the site of tumor formation in the Vector group, at which point all mice were euthanized by over-anesthetization. The tumors were then removed and their size measured. The results of the gross observations and extracted tumor sizes were added to the revised paper as Supplementary Figure S1, and relevant descriptions have been added to the text (Lines 237–243).

“Three weeks after transplantation, tumors were extracted for gross observation and measurement of mass size (Supplementary Figure S1). Ulcerations caused by an overgrowth of tumors were observed in tumors transplanted with parental SCCVII cells, vector, and CXCL10-expressing cell lines, while such tissue distraction was not observed in the CXCL9- and CXCL11-expressing cell lines. The size of the extracted tumors was significantly smaller in the CXCL9- and CXCL11-expressing cell lines as compared to the CXCL10-expressing cell line.”

  1. The results are simple. The antitumor mechanism should be investigated and appended.

Reply: The mechanism of the anti-tumor effects of chemokines was somewhat examined during this study. The anti-tumor effects of IFN-inducible chemokines have been shown to depend on the infiltration of CD8+ CTL and NK cells and their ability to suppress angiogenesis. As athymic nude mice lack T cells, we examined the expression of NK1.1 (CD161), a marker of NK cells, using IHC staining. However, no significant infiltration of NK1.1-positive cells was observed in CXCL9- and CXCL11-expressing tumor tissues. In addition, we investigated whether the expression of angiogenic factors, such as VEGF, was suppressed in CXCL9- and CXCL11-expressing tumor tissues using IHC. Although higher expression of VEGF was observed in tumor tissue transplanted with parental SCCVII cells, no significant suppression of VEGF expression was observed in CXCL9- and CXCL11-expressing tumor tissues. We are currently investigating the anti-tumor mechanisms of CXCL9- and CXCL11-expressing tumor tissues in nude mice.

We have revised the text in the Discussion (lines 336–348) and Conclusions (lines 359–360) as follows:

Discussion:

“The anti-tumor effects of IFN-inducible chemokines have been shown to depend on the infiltration of CD8+ CTL and NK cells [15–19] and their ability to suppress angiogenesis [12, 13]. Since athymic nude mice lack T cells, we examined the expression of NK1.1 (CD161), a marker of NK cells [51], using IHC staining in the course of this study. However, no marked infiltration of NK1.1-positive cells was observed in CXCL9- and CXCL11-expressing tumor tissues (data not shown). In addition, we investigated whether the expression of angiogenic factors, such as vascular endothelial growth factor (VEGF), was suppressed in CXCL9- and CXCL11-expressing tumor tissues using IHC. Although higher expression of VEGF was observed in tumor tissues transplanted with parental SCCVII cells, no significant suppression of VEGF expression was observed in CXCL9- and CXCL11-expressing tumor tissues (data not shown). Further studies are required to determine the anti-tumor mechanisms of CXCL9- and CXCL11-expressing tumor tissues in nude mice.”

Conclusions:

“Further studies are required to determine the anti-tumor mechanisms of CXCL9- and CXCL11-expressing tumor tissues, which are resistant to DPP4, to elucidate the immunosuppressive role of DPP4 in the tumor microenvironment, and to test whether DPP4 inhibitors restore the anti-tumor activity of IFN-inducible chemokines in an immunocompetent mouse model.”
